# Effects of Konjac Glucomannan on Oil Absorption and Safety Hazard Factor Formation of Fried Battered Fish Nuggets

**DOI:** 10.3390/foods11101437

**Published:** 2022-05-16

**Authors:** Jingwen Sun, Runlin Wu, Benlun Hu, Caihua Jia, Jianhua Rong, Shanbai Xiong, Ru Liu

**Affiliations:** 1College of Food Science and Technology, Huazhong Agricultural University, Wuhan 430070, China; sunjingwen1997@163.com (J.S.); wurunlin2018@163.com (R.W.); hubenlun1219@163.com (B.H.); chjia@mail.hzau.edu.cn (C.J.); rong@mail.hzau.edu.cn (J.R.); xiongsb@mail.hzau.edu.cn (S.X.); 2National R&D Branch Center for Conventional Freshwater Fish Processing (Wuhan), Wuhan 430070, China; 3Key Laboratory of Environment Correlative Dietology, Huazhong Agricultural University, Ministry of Education, Wuhan 430070, China

**Keywords:** konjac glucomannan, fried battered fish nuggets, oil content, advanced glycation end products, acrylamide

## Abstract

The purpose of this study was to investigate the effects of konjac glucomannan (KGM) on oil absorption and the formation of safety hazard factors in fried battered fish nuggets by measuring advanced glycation end products (AGEs) and acrylamide contents. Other physicochemical properties were determined to explore the reason for oil absorption and formation of safety hazard factors. The acrylamide was found mainly in the crust. The addition of 0.8% KGM could significantly reduce the acrylamide content (*p* < 0.05). For the battered sample, the AGEs content was far lower than the unbattered. The addition of 0.8% KGM could significantly reduce the AGEs content in the inner layer (*p* < 0.05). The microstructure showed that the sample with 0.8% KGM had the most compact crust. The compact crust reduced oil and malondialdehyde contents. Combined with the other indicators, the inhibitory effect of 0.8% KGM on acrylamide was closely related with the decreased extent of oil oxidation and Maillard reaction in the samples with 0.8% KGM. The inhibitory effect of 0.8% KGM on AGEs might originate from its lower oil content.

## 1. Introduction

Frying is one of the most common cooking methods for foods. With the acceleration of the pace of life, fried food has become the choice of more and more people because of its crispy taste, unique flavor and texture, affordable price, easy access and other characteristics [1,2,3]. However, fried foods are high in oil [4]. Frying might produce advanced glycation end products (AGEs) [5], acrylamide [6], 5-hydroxymethylfurfural [7] and other safety hazard factors which lead to a series of problems such as obesity and diabetes [8]. Therefore, reducing the oil and safety hazard factor contents of fried foods has become the focus of attention [9].

Direct deep frying can cause the product to shrink and the crust to harden [10]. Additionally, direct deep frying of potato products produces more acrylamide than batter frying [11]. The batter can effectively prevent the oil infiltration and water loss of food during frying [3]. However, current batter formulas are mainly a mixture of flour and starch [12]. Such fried foods contain higher oil and safety hazard factor contents in the batter layer [6]. Therefore, it is important to choose a suitable batter formula to replace the traditional batter. Currently, many studies have focused on adding functional ingredients to the batter to change the microstructure of the batter, thereby preventing the loss of water and the absorption of oil [13]. Hydrocolloids have gradually attracted people’s attention [14]. Xie et al. [15] found that adding guar gum, carrageenan, sodium alginate, xanthan gum and sodium carboxymethyl cellulose to the batter could reduce the oil content of fried battered fish nuggets. Jiang et al. [16] found that the contents of N^ε^-carboxymethyllysine and acrylamide in fish nuggets battered with chitosan, gum arabic and xanthan gum were significantly lower than those in directly fried fish nuggets.

KGM is a natural macromolecular heteropolysaccharide extracted from konjac tubers, which is composed of D-glucose and D-mannose through β-1,4-glycosidic bonds. It has excellent properties such as water-holding, thickening, film-forming and gelling properties [17]. Due to its good film-forming properties, it is also commonly used for the preservation of fruits and vegetables by film coating [18]. In terms of frying, adding KGM to the fish paste coating effectively prevented the penetration of oil into fish cakes [19]. However, few studies exist on the effect of the batter with KGM on oil and safety hazard factor contents of fried food.

In this study, grass carp was used as raw material. Fried battered fish nuggets were obtained with different proportions of KGM. The oil and safety hazard factor (AGEs, acrylamide) contents were determined. The aim was to explore the effects of KGM on the oil absorption and the formation of AGEs and acrylamide in fried battered fish nuggets. It was anticipated to provide a guidance and support for the development of healthy fried products.

## 2. Materials and Methods

### 2.1. Materials and Chemicals

Live grass carp (2.5–3.0 kg) was purchased from the Huazhong Agricultural University market (Wuhan, China). Arowana multipurpose wheat core wheat flour (Yihai Kerry Arowana Cereals and Oils Food Co., Ltd., Shanghai, China), MT200 modified starch (Tianjin Dingfeng Starch Development Co., Ltd., Tianjin, China), konjac glucomannan (food grade) (Hefei Shengrun Biological Products Co., Ltd., Hefei, China) and rapeseed oil (Shandong Luhua Group Co., Ltd., Yantai, China) were used. Salt and baking powder were purchased from Zhongbai Supermarket of Huazhong Agricultural University (Wuhan, China).

Petroleum ether (30–60 °C), HCl, chloroform, Sudan Red B, acetone, NaBH4, methanol, thiobarbituric acid, disodium EDTA, 1,1,3,3-tetraethoxypropane, formic acid, dichloromethane and hexane were purchased from Sinopharm Chemical Reagent Co., Ltd. (Shanghai, China). Carboxymethyl lysine (CML), carboxyethyl lysine (CEL) and methylglyoxalimidazolone (MG-H1) standards were purchased from Toronto Research Chemicals Inc. (Toronto, ON, Canada). Acrylamide standard was obtained from Sigma-Aldrich Co., Ltd. (St. Louis, MO, USA).

### 2.2. Preparation of Fried Battered Fish Nuggets

Live grass carp was killed, and the fish muscle was retained for further experiments. The fish muscle was cut into nuggets (3.0 × 3.0 × 1.5 cm) then salted with 1% NaCl for 6 h at 4 °C.

The battering recipe is shown in Table 1. Part of the wheat flour was replaced with KGM. Deionized water (batter:water = 1:1.2) was added to the batter to obtain a uniform batter. The battered fish nuggets were obtained by immersing the fish nuggets, which were precoated by modified starch, into the different batters then coated with wheat flour.

Two DF-6L fryers (Guangdong Jieguan Co., Ltd., Dongguan, China) were used to perform the frying experiment, which was carried out at 180 ± 2 °C as measured by an AT-380 infrared thermometer (Guang Dong Ma Co., Ltd., Dongguan, China). One fryer contained Sudan Red B (0.75 g/L) that was dissolved in rapeseed oil at 60 °C for 4 h and another fryer contained rapeseed oil only. In light of the effect of possible frying oil oxidation on the formation of safety hazard factors, fresh rapeseed oil was used each time.

Battered fish nuggets (6 pieces) were put into the fryer (containing 4 L rapeseed oil) and fried for 4 min each time. After frying, the samples were naturally cooled to room temperature. Next, the crust and inner layer of the battered fried fish nuggets were separated manually with a scalpel. All the samples were stored at −80 °C for further analysis. Among them, the samples fried in the fryer containing Sudan Red B were used for the Sudan Red staining experiment. The phenomenon of oil transport in the cross section of the sample was observed using a Stemi-508 stereo microscope (Zeiss, Oberkochen, German). Dyeing oil penetration depth was marked with Light tools 8.4 (Opinion Research Corporation Co., Ltd., Princeton, NJ, USA).

### 2.3. Measurement of Moisture and Oil Contents

The moisture and oil contents were determined, respectively, according to the AOAC official methods 950.46 and 960.39 [20]. The oil content was determined by an SIF-06A Soxhlet extractor (Jinan Alva Instrument Co., Ltd., Jinan, China). The samples (about 2.5~3.0 g) were extracted with petroleum ether for 6–8 h. The moisture and oil contents in the sample were expressed as g/g dry basis.

### 2.4. Measurement of Surface Oil, Penetrated Surface Oil and Structure Oil Contents

The surface oil (SO), penetrated surface oil (PSO) and structure oil (STO) were measured using the method of Bouchon et al. [21] with a slight modification. The battered fish nuggets were firstly fried in the rapeseed oil only fryer and then quickly (10 s before ending frying) moved to the Sudan Red B containing fryer. The fried battered fish nuggets were naturally cooled to room temperature and then weighed by a BS2102 electronic analytical balance (Zhejiang Baijie Instrument Co., Ltd., Hangzhou, China) as m (g). Petroleum ether was added in a dried beaker (m_1_, g). The naturally cooled fish nuggets were immersed in the petroleum-ether-containing beaker for 5 s, then the initially defatted fish nuggets were removed from the beaker followed by drying the beaker at 50 °C for 12 h to remove the petroleum ether. The beaker after drying was weighed and recorded as m_2_ (g). A Soxhlet extraction was performed on the fish nuggets that had been initially defatted, and the oil content of initially defatted fish nuggets was P. The oil extracted by the Soxhlet technique was diluted 20 times with petroleum ether and the absorbance value A was measured at 510 nm using a UV–vis spectrophotometer (Shimadzu, Kyoto, Japan).

In addition, the absorbances of 0.2, 0.4, 0.6, 0.8 and 1.0 g/L Sudan Red B standard solutions (in frying oil) were measured at 510 nm. The standard curve S between Sudan B concentration and absorbance value was established. The absorbance value A was substituted into the standard curve. The concentration of Sudan Red B corresponding to each sample was c_1_ (g/L). The SO, PSO and STO contents were calculated using the following equations:(1)$$P=\frac{\mathrm{Soxhlet}\;\mathrm{extraction}\;\mathrm{oil}\;\mathrm{quality}}{\;\mathrm{Initially}\;\mathrm{defatted}\;\mathrm{sample}\;\mathrm{quality}},$$
(2)$$\mathrm{SO}\left(\%\right)=\frac{\left(m_{2}-m_{1}\right)}{m}\times 100\%,$$
(3)$$\mathrm{PSO}\left(\%\right)=\left(\frac{c_{1}}{c_{0}}\right)\times P\times 100\%,$$
(4)$$\mathrm{STO}\left(\%\right)=\left(P-\mathrm{PSO}\right)\times 100\%,$$
c_0_ is the concentration of Sudan Red B in the dyed oil, which was 0.75 g/L. c_1_ refers to the concentration of Sudan Red staining solution in the Soxhlet extracted oil.

### 2.5. Measurement of Battering Rate

The fish nuggets before battering were weighed by a BL-2200H percent electronic balance (Zhejiang Baijie Instrument Co., Ltd., Hangzhou, China) as m_1_ (g). The battered fish nuggets were accurately weighed as m_2_ (g). The battering rate (B) was calculated using the following equation:(5)$$B\left(\%\right)=\frac{\left(m_{2}-{\;m}_{1}\right)}{m_{2}}\times 100\%.$$

### 2.6. Measurement of Malondialdehyde Content

According to the method of Jiang et al. [16], the malondialdehyde content was determined using a UV–vis spectrophotometer at 532 nm.

### 2.7. Microstructure Observation and Calculation of Pore Equivalent Diameter

The microstructure was observed using the method of Zeng et al. [22] with a slight modification. The crusts of the samples were cut into slices of about 2 mm × 2 mm × 1 mm with a ruler and a scalpel. The slices were fixed with 2.5% glutaraldehyde and dehydrated sequentially with gradient ethanol solutions for 10 min each. The slices were immersed for 30 min with isoamyl acetate. Finally, the slices were freeze-dried. The slices were glued to the stage using conductive glue and the surface was coated with gold film. Microstructural characteristics in the outer and inner surface of the slices were observed using a JSM-6390LV scanning electron microscope (Tianmei (China) Scientific Instrument Co., Ltd., Beijing, China) at 1000× and 3000× magnification.

According to the method of Zhu et al. [23], the electron microscope images were binarized by ImageJ 1.8.0 (National Institutes of Health, Bethesda, MD, USA) software. The pore equivalent diameter could represent the irregular pore size in the microstructure, and the calculation formula was as follows:(6)$$D\;\left(\mu m\right)=\sqrt{\frac{4A}{\pi }},$$
(7)$$A=\frac{S\times k}{n},$$

In the formula: D is the equivalent diameter of pores (μm), A is the average pore area (μm^2^), S is the picture area, n is the number of pores and k is the cumulative area percentage of pores.

### 2.8. Measurement of Nonfluorescent AGEs

The nonfluorescent AGEs contents were determined according to the methods of Niquet-Leridon and Tessier [24] and Jiang et al. [16], with a slight modification. N-hexane (3 mL) was added into 100 mg freeze-dried samples (freeze-dried using an FD-2A-100 frozen dryer (Boyikang Experimental Instrument Co., Ltd., Beijing, China)) and centrifuged at 4000 r/min for 15 min to defat. A borate buffer solution (2 mL, 0.2 mol/L, pH 9.2) and sodium borohydride solution (0.4 mL, 2 mol/L, dissolved in 0.1 mol/L NaOH solution) were added to the defatted samples, and reacted at 4 °C for 8 h. Hydrochloric acid solution (4 mL, 6 mol/L) was added into the reacted protein precipitate, and acidated at 110 °C for 24 h. The reacted hydrolyzate of hydrochloric acid (1 mL) was dried in a DHG-9240A oven (Shanghai Zhuohao Laboratory Equipment Co., Ltd., Shanghai, China) at 60 °C, then redissolved in ultrapure water (1 mL) and filtered through a 0.22 μm membrane. The filtered solution was passed through an MCX solid phase extraction column (Thermo Fisher Co., Ltd., Waltham, MA, USA). The eluate was evaporated and dissolved in 1 mL of ultrapure water. Finally, the solution was filtered (0.22 μm nylon) and subjected to chromatographic separation.

The CML, CEL and MG-H1 contents of fried battered fish nuggets were determined by ultraperformance liquid chromatography (Waters Co., Ltd., Milford, MA, USA) and detected in multiple reaction monitoring mode using a Xevo TQ MS (Waters Co., Ltd., Milford, MA, USA). The mobile phase consisted of an aqueous solution containing 5 mM ammonium acetate and 0.1% formic acid as solvent A, and 100% acetonitrile as solvent B. Separations were performed using a linear gradient of A into B at a flow rate of 300 μL/min. The source temperature was set to 110 °C and the desolvation temperature was 350 °C. The mass spectrometer was operated in positive mode ESI with a capillary voltage of 4 kV. The linear range of calibration curves were 0.01~0.2 μg/mL and the quantification of samples was achieved by measuring their peak area ratio and comparing with the external standard curves.

### 2.9. Measurement of Fluorescent AGEs

Phosphate buffer (50 mmol/L, pH 7.4) was added into the sample with a ratio of 1:10 (g/mL, sample: phosphate buffer). Then, the mixture was stirred at 37 °C for 1 h. Finally, the reactant was centrifuged at 4000 r/min for 5 min and the supernatant was collected. Fluorescence values were measured using an F-4600 fluorescence spectrophotometer (Shanghai Zhuohao Laboratory Equipment Co., Ltd., Shanghai, China).

### 2.10. Measurement of Acrylamide Content

The acrylamide content was measured according to the method of Michalak et al. [25] with a slight modification. The sample (4 g) was weighed and defatted with n-hexane (20 mL). NaCl solution (10 mL, 3 mol/L) and formic acid solution (1 ml, 0.1%) was added into the defatted sample. The reactant was followed by magnetic stirring for 20 min and centrifugation at 10,000 r/min for 15 min. The supernatant was collected. Then, ethyl acetate (20 mL) was added to the supernatant, and the upper solution was collected by shaking for 5 min. The operation was repeated twice. The obtained extraction solution was evaporated using a rotary evaporation and then dried with nitrogen gas at 50 °C. Then, the solution was dissolved in ultrapure water and was injected into the sample bottle through a 0.45 μm water filtration membrane. A Hypercarb column (4.6 × 100 mm, 3.0 μm, Thermo Fisher Scientific Co., Ltd., Shanghai, China) was used for the e2695 high-performance liquid chromatographic system (Waters Co., Ltd., Milford, MA, USA). The sample and acrylamide standard were eluted at 25 °C using 95% ultrapure water and 5% methanol at a flow rate of 0.8 mL/min as the mobile phase. The standard curve was made between the peak area and concentration of the acrylamide standard. The acrylamide content was quantified according to the peak area of acrylamide.

### 2.11. Statistical Analysis

All experiments were run in triplicate. Excel 2019 (Microsoft, Redmond, WA, USA) was used for data statistics. Origin 2020 (OriginLab, Co., Ltd., Northampton, MA, USA) was used for graphing. SPSS 25.0 (International Business Machines Corporation, Armonk, NY, USA) was used for the significance analysis (*p* < 0.05 indicated a significant difference).

## 3. Results and Discussion

### 3.1. Moisture and Oil Contents of Fried Battered Fish Nuggets

The moisture and oil contents of fried battered fish nuggets with different proportions of KGM are shown in Figure 1A,B. The addition of 0.4% and 0.8% KGM significantly increased the moisture content of the crust and reduced the oil content of the crust and the inner layer (*p* < 0.05). When the addition amount of KGM exceeded 0.8%, the moisture content of the product significantly decreased (*p* < 0.05) and the oil content significantly increased (*p* < 0.05). Zhang et al. [26] found that KGM would expand rapidly and caused conformational changes after water absorption, so that the crust had a high water-holding capacity. Therefore, we thought the increased moisture content might be due to the enhanced water-holding capacity of the crust after adding KGM. However, when the addition amount of KGM exceeded 0.8%, it was most likely that most of the free water in the batter was absorbed by KGM, which decreased the dispersion of wheat flour, starch and KGM in the batter, resulting in the batter being uneven and coarse. According to Rahimi and Ngadi [27], the surface irregularities and rough structure of the batter could lead to additional water loss and oil absorption. The samples with high moisture content all had lower oil content, which might be related to the moisture displacement mechanism [28]. According to this mechanism, water is first evaporated from the food during frying, and then the oil penetrates into the void left by the evaporation of the water [28]. It could explain the negative correlation between moisture and oil contents.

The moisture content of the inner layer was significantly higher than that of the crust (*p* < 0.05), and it was the opposite for the oil content. It was most likely that the crust acted as a barrier to hinder the loss of water in the inner layer and the penetration of the outer oil.

### 3.2. Battering Rate of Fried Battered Fish Nuggets

The battering rate affected the quality and yield of the product and was directly related to the viscosity of the batter [15]. Figure 1C shows the battering rate of fried battered fish nuggets with different proportions of KGM. Samples with 0.8–1.6% KGM had significantly higher battering rate than samples with 0–0.4% KGM (*p* < 0.05). KGM could absorb water 80–100 times its own weight, and the thick of its aqueous solution increased with the increase of KGM addition. [29]. It was most likely that a higher KGM addition could absorb more free water, making the batter difficult to drip, thereby increasing battering rate and product yield.

### 3.3. Oil Distribution of Fried Battered Fish Nuggets

Figure 2A shows the oil penetration of fried battered fish nuggets with different proportions of KGM. The position of the black curve is the boundary between the crust and the inner layer. The red area represents the penetration depth of the Sudan Red dyed oil into the product. Frying caused the batter to lose water and shrink to form a crust. A thicker crust was closely related to a higher battering rate with increasing KGM additions. Dyeing oil was mainly present in the crust. The crust prevented the oil from penetrating into the fish nuggets. There was no significant difference in the contents of SO and STO between all samples (*p* < 0.05). The PSO content decreased followed by an increase with increasing KGM additions and reached a minimum value when the KGM addition was 0.8%. As shown in Figure 2C, the oil penetration depth decreased followed by an increase with the increase of KGM additions and reached a minimum value when the KGM addition was 0.8%. PSO mainly penetrated into the product in the cooling stage along the pores generated during frying, due to the internal and external pressure difference and capillary force [14]. The Sudan Red staining chart could intuitively reflect the content of PSO in the sample [14]. The higher the moisture content of the sample and the compacter the crust, the lower the PSO content [20]. With both 0.4% and 0.8% of added KGM, the moisture content was high. However, only the samples with 0.8% KGM had the lowest PSO content. It was speculated that this was related to the compactness of the crust, which would be analyzed in combination with the microscopic topography of the crust.

STO indicated the absorbed oil during frying, which reacted with ingredients in the food [21]. As shown in Figure 2B, the STO content was lower than the PSO and SO contents in the same sample. It was shown that the oil content absorbed during frying was significantly lower than the one absorbed during cooling (*p* < 0.05). This was consistent with the results of Garmakhani et al. [30] and Zhang et al. [31].

### 3.4. Microstructures of the Crust of Fried Battered Fish Nuggets

The microstructures of the crust with different proportions of KGM are shown in Figure 3A, and pore equivalent diameters are shown in Figure 3B. The pore equivalent diameter could reflect the size of the pores in the structure [22]. The inner surface of the crust was smoother than the outer surface. The size and number of pores in the inner surface depended on the KGM addition (Figure 3A). For the crust with 0.8% KGM, the inner surface was the most compact and smooth and obtained the minimum pore equivalent diameter. It was most likely that the addition of 0.8% KGM effectively prevented the penetration of oil and further reduced the oil and PSO contents (Figure 1B and Figure 2B). When the KGM addition exceeded 0.8%, the crust structure became rough and bigger pores were formed (Figure 3B). Therefore, the oil infiltration increased, resulting in an increase in the oil and PSO contents. Therefore, we thought that 0.8% was the appropriate addition amount of KGM in this experiment. Lumanlan et al. [14] found that adding an appropriate amount of hydrocolloids such as 0.5% xanthan gum and 0.5% guar gum into the batter would form a smooth and compact crust to prevent oil from infiltrating.

### 3.5. Malondialdehyde Content of Fried Battered Fish Nuggets

Figure 4 shows the malondialdehyde content of fried battered fish nuggets with different proportions of KGM. Malondialdehyde is a secondary oxidation product of oil, which can characterize the degree of oil oxidation [32]. As the amount of KGM increased, the malondialdehyde content decreased followed by an increase. It reached a minimum value when the KGM addition was 0.8%. The batter itself did not contain oil, so the malondialdehyde in the fried battered fish nuggets mainly came from the frying oil. The compact crust formed by the batter with 0.8% KGM prevented the penetration of oil and further reduced the malondialdehyde content. This indicated that 0.8% KGM could slow down the process of oil oxidation. The high malondialdehyde content of the samples with 1.2% and 1.6% KGM was closely related with their high oil content.

The malondialdehyde content in the inner layer of the fried fish nuggets was significantly lower than that of the crust (*p* < 0.05). On the one hand, the lower temperature of the inner layer of the fried battered fish nuggets weakened the degree of oil oxidation; on the other hand, the amount of oxygen and oil of the inner layer was very low, which also depressed the malondialdehyde content.

### 3.6. AGEs of Fried Fish Nuggets

According to the above research results, the batter with 0.8% KGM was chosen to study the effect of KGM on the safety hazard factors (AGEs and acrylamide). Figure 5A–D shows the nonfluorescent AGEs and fluorescent AGEs contents of fried battered fish nuggets. Carboxymethyl lysine (CML), carboxyethyl lysine (CEL) and methylglyoxalimidazolone (MG-H1) are three typical representative products of nonfluorescent AGEs [33,34]. Battering significantly reduced the contents of AGEs in the crust and the inner layer of fried fish nuggets (*p* < 0.05). There was no significant difference in the contents of AGEs in the crust between the sample with 0% and 0.8% KGM (*p* < 0.05), but the AGEs content in the inner layer of the sample with 0.8% KGM was significantly lower than the sample without KGM (*p* < 0.05). Fish nuggets were rich in protein. When the fish nuggets directly contacted the frying oil, the Maillard reaction was severe at high temperature, resulting in the formation of more AGEs. Compared with the unbattered sample, the decreased AGEs contents in the crust of the battered sample might originate from their low protein content. The main components of the batter were flour and starch, and the protein contents were far less than that of fish nuggets. The number of amino acid residues that could be modified by glycosylation in the batter was reduced, thereby depressing the formation of AGEs [35]. Compared with the unbattered sample, the decreased AGEs contents in the inner layer after battering might have originated from the low oil content. The lower the oil content, the lower the carbonyl compound content [36]. Carbonyl compounds are important precursors for the formation of AGEs, and the decrease in carbonyl compound contents can reduce the contents of AGEs [37]. Additionally, 0.8% KGM significantly decreased the AGEs contents in the inner layer of the sample (*p* < 0.05). It is possible that the compact crust prevented the frying oil from infiltrating, further reducing the contents of carbonyl compound produced through the oil oxidation pathway.

The CML and CEL contents in the crust of the unbattered sample were significantly lower than those in the inner layer (*p* < 0.05), which might be related with the higher protein oxidation in the crust. Zhu et al. [38] found that mild protein oxidation could promote the production of CML and CEL, and when the degree of protein oxidation was deepened, the production of CML and CEL would be inhibited. The MG-H1 content in the crust of the battered sample was significantly higher than that in the inner layer, while the CML and CEL contents were opposite (*p* < 0.05). The difference might have originated from the different types and contents of amino acids. The precursor amino acid of CML and CEL is lysine, while the precursor amino acid of MG-H1 is arginine [32]. After battering, the crust was mainly composed of wheat flour and starch, while the inner layer was fish muscle. The arginine content was high in the wheat flour and starch, and the lysine content was high in the fish muscle [39,40]. The fluorescent AGEs content in the crust was significantly higher than that in the inner layer (*p* < 0.05). It was most likely that the higher fluorescent AGEs originated from the higher temperature, the higher oil content and the lower moisture content of the crust. Trevisan et al. [41] reported that the contents of fluorescent compounds increased with decreasing moisture content and increased with increasing temperature. The higher the oil content, the more AGEs were produced through the oil oxidation [36,37].

### 3.7. Acrylamide Content of Fried Battered Fish Nuggets

The acrylamide content in the unbattered sample and the inner layer of the battered samples was lower than the tested value. Figure 5E shows the acrylamide content in the crust of fried battered fish nuggets with KGM. A 0.8% KGM addition significantly decreased the acrylamide content (*p* < 0.05). The higher acrylamide content in the crust was mainly related to the higher starch and wheat flour contents. Acrylamide was mainly formed by the reaction of asparagine and carbonyl. The main sources of carbonyl were reducing sugar and oil oxidation. The starch and wheat flour in the crust contained more reducing sugars than the fish muscle [42]. The decreased acrylamide content in the crust with 0.8% KGM might be related with its lower degree of oil oxidation and higher moisture content. The lower degree of oil oxidation resulted in less acrylamide content [43,44]. A high moisture content could inhibit the Maillard reaction [35].

### 3.8. Discussion

AGEs were produced by carbonyl compounds and proteins through a Maillard reaction. Oil oxidation could also provide carbonyl compounds, which could promote the formation of AGEs [45]. Acrylamide was mainly produced by a Maillard reaction between asparagine and carbonyl compounds. Acrolein/acrylic acid produced by oil oxidation could also generate acrylamide in the presence of ammonia [43,44]. The batter with 0.8% KGM formed a compact crust, which could prevent the water loss and oil absorption. Battering could significantly decreased AGEs contents in the crust because of the low protein contents (*p* < 0.05). The addition of 0.8% KGM in the batter could further significantly reduce the AGEs content in the inner layer and acrylamide content in the crust (*p* < 0.05). The reason might originate from the fact that the compact crust with 0.8% KGM slowed down the degree of the Maillard reaction and oil oxidation, thereby reducing the formation of acrolein, acrylic acid and carbonyl compounds, which reduced the AGEs and acrylamide contents.

Acrylamide is mainly produced in starchy foods [44]. This is because the reducing sugar contents in protein foods are low, which limit the production of acrylamide [42]. Additionally, the amino acids contained in proteins can easily react with the unsaturated double bonds in acrylamide to eliminate acrylamide [45]. This was also the reason why acrylamide was undetectable in the unbattered sample and the inner layer of the fried battered samples. In addition, hydrophilic groups contained in hydrocolloids could interfere with the decarboxylation process or react with some acrylamide intermediates, resulting in the reduction of acrylamide content [34]. Therefore, the decrease of acrylamide content after adding KGM might also be related with the hydrophilic groups, which needs to be further confirmed.

## 4. Conclusions

Battering prevented the fish nuggets from directly contacting frying oil, thereby effectively reducing the AGEs content in the fried fish nuggets. The addition of 0.8% KGM in the batter could further significantly reduce the AGEs content in the inner layer (*p* < 0.05). The CML and CEL contents in the inner layer were significantly higher than those in the crust, while for the MG-H1 and fluorescent AGEs contents, it was the opposite (*p* < 0.05). This difference might be related with the different types and quantities of amino acids. The acrylamide was mainly found in the crust. The addition of 0.8% KGM could significantly reduce the acrylamide content (*p* < 0.05). The compact crust was formed after adding 0.8% KGM to the batter, which significantly reduced the water loss, oil and PSO contents of fried battered nuggets (*p* < 0.05). Additionally, 0.8% KGM slowed down the degree of the Maillard reaction and oil oxidation and therefore decreased the AGEs and acrylamide contents. Based on the experimental results, 0.8% KGM could be added in batter in order to depress oil absorption and the formation of AGEs and acrylamide in fried battered foods.

## Figures and Tables

**Figure 1 foods-11-01437-f001:**
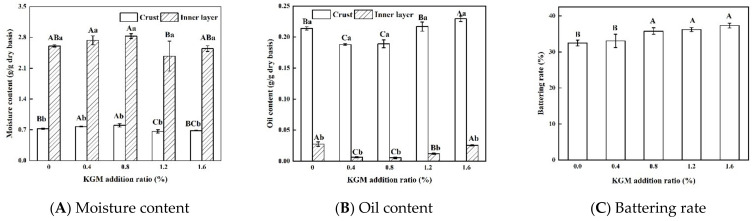
Moisture content (**A**), oil content (**B**), and battering rate (**C**) of fried battered fish nuggets with different proportions of KGM. Note: Different capital letters indicate that there are significant differences between different conditions (*p* < 0.05). Different lowercase letters indicate that there are significant differences between the crust and the inner layer in the same condition (*p* < 0.05).

**Figure 2 foods-11-01437-f002:**
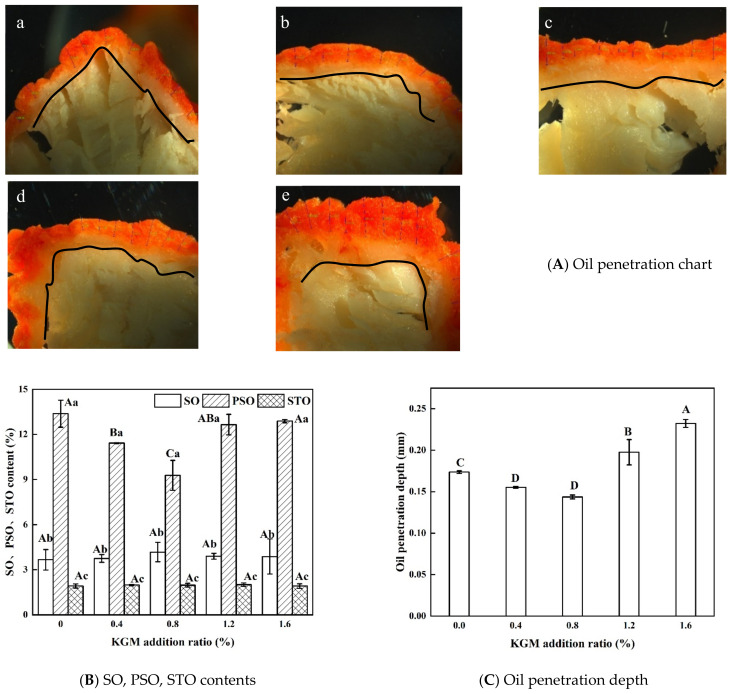
Oil distribution of fried battered fish nuggets with different proportions of KGM. (**A**) Indicates the Sudan Red staining of sample. (a–e) indicate the images of samples under 0% KGM, 0.4% KGM, 0.8% KGM, 1.2% KGM and 1.6% KGM, respectively. (**B**) indicates the SO, PSO, STO contents of sample. (**C**) indicates the oil penetration depth of sample. Note: Different capital letters indicate that there are significant differences between different conditions (*p* < 0.05). Different lowercase letters indicate that there are significant differences between the crust and the inner layer in the same condition (*p* < 0.05).

**Figure 3 foods-11-01437-f003:**
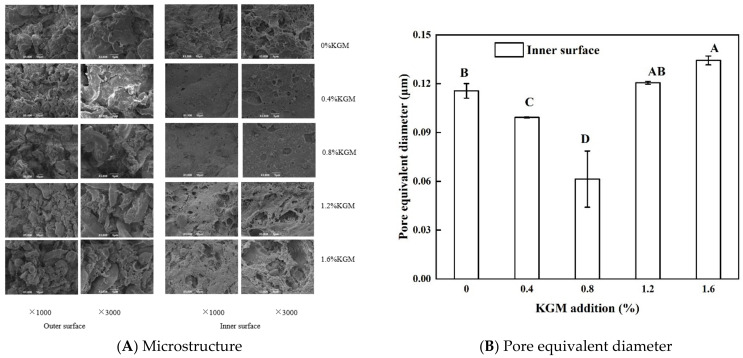
Microstructure (**A**) and the pore equivalent diameter (**B**) of the crust with different proportions of KGM. Note: different capital letters indicate that there are significant differences between different conditions (*p* < 0.05).

**Figure 4 foods-11-01437-f004:**
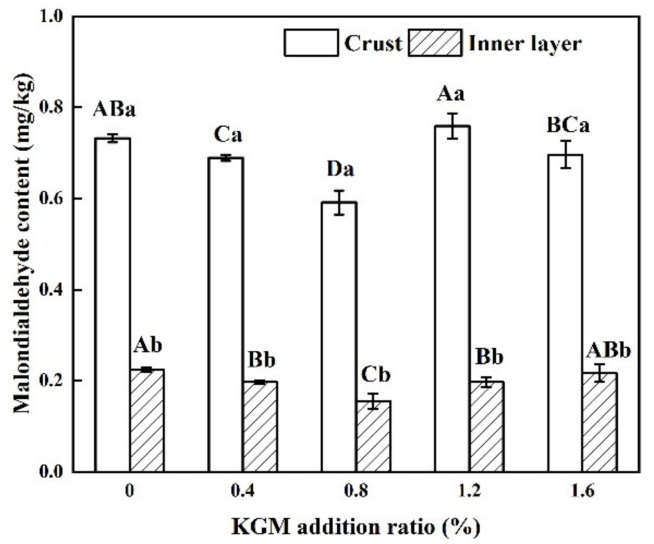
Malondialdehyde content of fried battered fish nuggets with different proportions of KGM. Note: Different capital letters indicate that there are significant differences between different conditions (*p* < 0.05). Different lowercase letters indicate that there are significant differences between the crust and the inner layer in the same condition (*p* < 0.05).

**Figure 5 foods-11-01437-f005:**
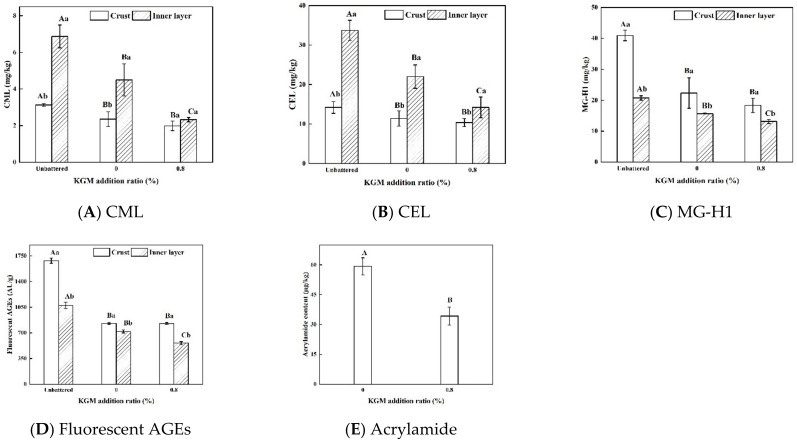
Nonfluorescent, fluorescent AGEs and acrylamide contents of fried fish nuggets with KGM. Note: Different capital letters indicate that there are significant differences between different conditions (*p* < 0.05). Different lowercase letters indicate that there are significant differences between the crust and the inner layer in the same condition (*p* < 0.05).

**Table 1 foods-11-01437-t001:** Battering Recipe.

KGM Addition Ratio (%)	Wheat Flour (%)	Modified Starch (%)	Baking Powder (%)	Salt (%)	KGM (%)
0	58	40	0.5	1.5	0
0.4	57.6	40	0.5	1.5	0.4
0.8	57.2	40	0.5	1.5	0.8
1.2	56.8	40	0.5	1.5	1.2
1.6	56.4	40	0.5	1.5	1.6

## Data Availability

Not applicable.

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
