# Peer review of "Effects of Konjac Glucomannan on Oil Absorption and Safety Hazard Factor Formation of Fried Battered Fish Nuggets"

_foods, 2022, doi:10.3390/foods11101437_

Round 1

Reviewer 1 Report

This manuscript presents results on the effects of Konjac glucomannan (KGM) on fat reduction and oil oxidation of fried fish products. Moreover, fat penetration was visualized by using dyed oil, and microstructures were investigated as well. The information may be useful for future research in fat reduction for similar products. However, the writing needs much improvement by a native English speaker. This reviewer had a difficult time understanding what the authors tried to say. Also, the originality of the paper and the scientific significance/novelty is not given with the presented results. Similar research was published in which Supawong et al. (2018) proposed an identical coating material (Konjac glucomannan) for a similar fish product (i.e., surimi).

The followings are specific comments for the authors to consider in the revision.

  1. L25, L33 described the same thing; Also, frying is a common cooking method for foods or a common food preparation method, instead of a food processing method.
  2. L56, please specify the objectives of this study
  3. L62, it is not clear how did authors prepare fish? Fish was fresh or frozen?
  4. L70, what is the definition of fish nugget vs. fish cake? Please include the geometry and dimension of the fish product to be fried.
  5. L74-75: it is not a sentence. There are many similar language/grammar issues throughout the text (e.g., L76, L109, L152 etc.). Authors need to carefully revise all of them.
  6. L80 Table 1: I suggest authors include the total mass or express each ingredient in percentage (%).
  7. L82: 105C drying was used to measure the moisture contents of samples. But I concerned that such a high temperature would cause lipid oxidation or evaporation of other heat-sensitive compounds, thus giving a misleading result. I assume that the authors did moisture measurements first and then oil extraction. If in this case, the authors were not able to get accurate fat content of samples already exposed to high-temperature drying. The reference cited was not included in the reference list, and it is not an international standard. I suggest authors refer to AOAC standards (Official Method 926.12: Moisture and volatile matter in oils and fats; Official Method 948.22: Fat (crude) in nuts and nut products). Vacuum drying at a low temperature should be used.
  8. L83: Please specify the solvent, extraction time the mass of samples in the Soxhlet.
  9. Section 2.3 was too rough to believe. If the measurement method for moisture and fat was not correct, the results of this study would be totally misleading!
  10. L92: “Petroleum ether was added in a beaker (m1) that had been dried to constant weight.” It is very difficult to understand this sentence.
  11. L93: Cooled? The fired fish samples were allowed to cool to room temperature? Or by other water/ice cooling methods?
  12. (1)-(5): Please include units. If it is a weight, please include what type of balance was used.
  13. L115: How did the authors cut the sample to exact 2mm*2mm*1mm?
  14. Please bear in mind that every time you use the word “significantly”, you need to specify the p-value or p<? (L14, L15, L181, L184, etc.)
  15. L185: include the reference to support your statement.
  16. Be sure that conclusions are always supported by the given data (e.g. L188; L209: You did not determine the free water. Therefore, I would suggest: It was assumed/was most likely).
  17. 1: what is the unit of MC (dry basis. or wet basis)? What is unit of fat content? Why authors expressed them in percentage?
  18. 2: How did the authors draw a back curve? What sort of image process technology was used to distinguish the crust and the interior?
  19. 6: What is its purpose? No data was presented. It is more like a graphical abstract.

Overall, the manuscript needs major revisions to improve clarity and logic flow. I hope authors pay more attention to Material & Method which is not acceptable in the present version.

Author Response

Dear Reviewer,

Thank you for reviewing our manuscript and constructive suggestions concerning our manuscript entitled “Effects of Konjac glucomannan on oil absorption and safety hazard factor formation of fried battered fish nuggets” to Foods (ID: foods-1703084). These suggestions would help us to improve the quality of the paper, as well as the important guiding significance to our researches. We have studied comments carefully and have made correction. Revised portion are marked in red in the paper. The main corrections in the paper and the response to the reviewer’s comments are as follows.

Response to the reviewer’s comments,

Q1: L25, L33 described the same thing; Also, frying is a common cooking method for foods or a common food preparation method, instead of a food processing method.

Answer: Thanks for the valuable suggestion. “Frying is one of the most commonly used methods in food processing.” was replaced by “Frying is one of the most common cooking methods for foods.”.  “Direct frying and batter frying are common food processing methods.” had been deleted.

Q2: L56, please specify the objectives of this study.

Answer: Thanks for the valuable suggestion.  “The aim was to explore effects of KGM on the oil absorption and the formation of AGEs and acrylamide of fried battered fish nuggets.” was added in the revision.

Q3: L62, it is not clear how did authors prepare fish? Fish was fresh or frozen?

Answer: Thanks for the valuable suggestion. Live grass carp (2.5-3.0 kg) was prepared from the Huazhong Agricultural University market. Live grass carp was killed and the fish muscle was retained for further experiments. The fish muscle was cut into nuggets (3.0×3.0×1.5 cm) followed by salted with 1% NaCl for 6h at 4°C.

Q4: L70, what is the definition of fish nugget vs. fish cake? Please include the geometry and dimension of the fish product to be fried.

Answer: Thanks for the suggestion. The fish nuggets are cut directly from the fish muscle. Fish cake is a surimi product. It is made by pouring surimi into mold followed by heating. The fried battered fish nuggets were 3.0×3.0×1.5 cm, which was added in the revision.

Q5: L74-75: it is not a sentence. There are many similar language/grammar issues throughout the text (e.g., L76, L109, L152 etc.). Authors need to carefully revise all of them.

Answer: Thank you very much. It was modified by “The battered fish nuggets were obtained by immersing the fish nuggets, which were pre-coated by modified starch, into the different batter followed by coated with wheat flour.”. The other sentences were replaced by “The battered fish nuggets was accurately weighed as m2 (g)”, “Phosphate buffer (50mmol/L, pH 7.4) was added into the sample with a ratio of 1:10 (g/mL, sample: phosphate buffer).” And we carefully checked and modified other grammatically incorrect sentences and marked them in red in the revision.

Q6: L80 Table 1: I suggest authors include the total mass or express each ingredient in percentage (%).

Answer: Thanks for your suggestion. We express each ingredient in percentage (%).

Q7: L82: 105C drying was used to measure the moisture contents of samples. But I concerned that such a high temperature would cause lipid oxidation or evaporation of other heat-sensitive compounds, thus giving a misleading result. I assume that the authors did moisture measurements first and then oil extraction. If in this case, the authors were not able to get accurate fat content of samples already exposed to high-temperature drying. The reference cited was not included in the reference list, and it is not an international standard. I suggest authors refer to AOAC standards (Official Method 926.12: Moisture and volatile matter in oils and fats; Official Method 948.22: Fat (crude) in nuts and nut products). Vacuum drying at a low temperature should be used.

Answer: Thanks for the valuable suggestion. I am so sorry that the method of moisture and oil contents is not clear. Your recommended AOAC 926.12 method is vacuum drying. The method for measuring moisture in our experiments is AOAC 950.46. This method measures moisture by drying in an oven at 105°C. According to the literature [Khazaei et al. 2015, Zeng et al. 2016], the measurement of the moisture content of fried food is basically selected at 105°C. The boiling point of rapeseed oil is very high (335℃), and reactions such as decomposition, evaporation will not occur at 105 °C. A small amount of oil may be oxidized, but it hardly influences the overall trend. The moisture and oil contents were measured in order to explore the effect of adding different proportions of KGM, so the method AOAC 950.46 can be selected. Method AOAC 948.22 is usually used to measure the oil content of nuts. Our experiment object was fried battered fish nuggets, which belonged to the meat category and was more suitable for measurement by AOAC 960.39. The moisture content was expressed on wet basis, the oil content was expressed on dry basis. This was to exclude the interference of moisture on the determination of oil content. The moisture and oil contents measurement methods mentioned in the manuscript were national standards GB5009.3-2016 and GB5009.6-2016. These two methods are consistent with international standards AOAC 950.46 and 960.39. Section 2.3 had been modified in the revision. Thank you again.

Reference:

Khazaei N, et al. Effect of active edible coatings made by basil seed gum and thymol on oil uptake and oxidation in shrimp during deep-fat frying. J. Sci Food Agri., 2015, 137:249-254.

Zeng H et al. Reduction of the fat content of battered and breaded fish balls during deep-fat frying using fermented bamboo shoot dietary fiber. LWT-Food Sci. Technol. 2016, 73, 425-431.

Q8: L83: Please specify the solvent, extraction time the mass of samples in the Soxhlet.

Answer: Thanks for the valuable suggestion. The samples (about 2.5~3.0 g) were extracted with petroleum ether for 6-8 h. It was added in the revision.

Q9: Section 2.3 was too rough to believe. If the measurement method for moisture and fat was not correct, the results of this study would be totally misleading!

Answer: Thanks for the valuable suggestion. Section 2.3 had been modified in the revision to “According to the AOAC official methods 950.46 and 960.39 [21], the moisture and oil contents were determined respectively. The oil content was determined by SIF-06A Soxhlet Extractor (Jinan Alva Instrument Co., Ltd., Jinan, Shandong, China). The samples (about 2.5~3.0 g) were extracted with petroleum ether for 6-8 h. The moisture content was expressed on wet basis, the oil content was expressed on dry basis.”.

Q10: L92: “Petroleum ether was added in a beaker (m1) that had been dried to constant weight.” It is very difficult to understand this sentence.

Answer: I am sorry for the poor expression. It was replaced by “Petroleum ether was added in a dried beaker (m1, g).”.

Q11: L93: Cooled? The fired fish samples were allowed to cool to room temperature? Or by other water/ice cooling methods?

Answer: Thanks for the suggestion. The fried battered fish nuggets were naturally cooled to room temperature. “The fried battered fish nuggets were naturally cooled to room temperature and then weighed by BS2102-electronic analytical balance (Zhejiang Baijie Instrument Co., Ltd., Zhejiang, China) as m (g).” was added in the revision.

Q12: (1)-(5): Please include units. If it is a weight, please include what type of balance was used.

Answer: Thanks for the suggestion. Equations 1-5 have all added units in the revision. “BS2102-electronic analytical balance (Zhejiang Baijie Instrument Co., Ltd., Zhejiang, China)” was added in the revision.

Q13: L115: How did the authors cut the sample to exact 2mm*2mm*1mm?

Answer: Thanks for the suggestion. We cut the sample with a ruler and scalpel. It was replaced by “The crusts of the samples were cut into slices of about 2 mm x 2 mm x 1 mm with a ruler and a scalpel”.

Q14: Please bear in mind that every time you use the word “significantly”, you need to specify the p-value or p<? (L14, L15, L181, L184, etc.)

Answer: Thanks for the valuable suggestion. p<0.05 was all added in the revision.

Q15: L185: include the reference to support your statement.

Answer: Thanks for the suggestion. It was replaced by “Zhang et al. [27] found that KGM would expand rapidly and caused conformational changes after water absorption, so that the crust had a high water-holding capacity. So, we thought the increased moisture content might be due to the enhanced water-holding capacity of the crust after adding KGM.”.

Q16: Be sure that conclusions are always supported by the given data (e.g. L188; L209: You did not determine the free water. Therefore, I would suggest: It was assumed/was most likely).

Answer: Thanks for the valuable suggestion. It was modified as follows: “However, when the addition amount of KGM exceeded 0.8%, it was most likely that most of the free water in the batter was absorbed by KGM, which decreased the dispersion of wheat flour, starch and KGM in the batter, resulting in the batter uneven and coarse.”. “It was most likely that higher KGM addition could absorb more free water, making the batter difficult to drip, thereby increasing battering rate and product yield.”.

Q17: 1: what is the unit of MC (dry basis. or wet basis)? What is unit of fat content? Why authors expressed them in percentage?

Answer: Thanks for the suggestion. “The moisture content was expressed on wet basis, the oil content was expressed on dry basis.” was added in the 2.3. Measurement of moisture and oil contents. The unit of oil content was percentage. There are two reasons: 1) We had referenced a lot of literature, basically choosing percentage as the unit of oil content. Reference:

Khazaei N, et al. Effect of active edible coatings made by basil seed gum and thymol on oil uptake and oxidation in shrimp during deep-fat frying. J. Sci. Food Agri. 2015, 137:249-254.

Shen SD, et al. Effect of batter formula on qualities of deep-fat and microwave fried fish nuggets. J. Food Eng. 2009, 95(2):359-364.

2) It is more convenient to compare and calculate between samples.

Q18: 2: How did the authors draw a back curve? What sort of image process technology was used to distinguish the crust and the interior?

Answer: Thanks for the suggestion. The crust of our sample was batter, and the inner layer was fish meat. After frying, the sample would shrink due to moisture loss, resulting in obvious separation of the crust and the inner layer. So, it could be clearly seen the demarcation between the crust and the inner layer. We drew the curves using PowerPoint.

Q19: 6: What is its purpose? No data was presented. It is more like a graphical abstract.

Answer: Thanks for the suggestion. Figure 6 was mainly a summary of the full text. After careful discussion, we thought figure 6 was more like a graphical abstract. Therefore, we decided to delete it. Thank you again.

Reviewer 2 Report

Dear authors,

The manuscript is quite interesting and will attract a large readership. The comments are below as well as in the manuscript attached.

Line 12 – AGEs – define at first mention

Methods should be written as a reported speech or in the past tense. "The fish was weighed..."(lines71-76, 109, 130, 136, 146, 152, 159,

Line 88 – The name of the author is ‘Bouchon’ and not ‘Ouchon’

Line 95 and 97 – bold or italicise.

 Equation 3 – Define C1

Line 125 - This should come before the frying process

Line 142 – baked or fried?

Line 212 – oil penetration chart not mentioned in the Figure legend.

Line 239 - the one absorbed during cooling

Lines 248-249: If the inner surface was smooth as seen in the microstructure, how come the pore equivalent diameter is higher? One would think the rougher outer surface should possess a higher pore equivalent diameter. But that is not the case in the results. Please explain.

Line 256 - The statement is not clear. Is this a proposed solution to counter the effect of oil increase with increasing KGM?

Line 283 - Which previous research? Please cite.

Figures 5b and c – the standard deviation is quite high meaning the triplicate data were spread out rather than clustered as other results. The coefficient of variation for the two datasets should be determined. If it is less than 20%, the data should be accepted. If not, the data should be re-calculated.

Author Response

Dear Reviewer,

Thank you for reviewing our manuscript and constructive suggestions concerning our manuscript entitled “Effects of Konjac glucomannan on oil absorption and safety hazard factor formation of fried battered fish nuggets” to Foods (ID: foods-1703084). These suggestions would help us to improve the quality of the paper, as well as the important guiding significance to our researches. We have studied comments carefully and have made correction. Revised portion are marked in red in the paper. The main corrections in the paper and the response to the reviewer’s comments are as follows.

Response to the reviewer’s comments,

Q1: Line 12 – AGEs – define at first mention

Answer: Thanks for your suggestion. The definition had been added in the Line14.

Q2: Methods should be written as a reported speech or in the past tense. "The fish was weighed..."(lines71-76, 109, 130, 136, 146, 152, 159,

Answer: Thanks for your suggestion. We carefully checked and modified these grammatically incorrect sentences and marked them in red in the revision.

Q3: Line 88 – The name of the author is ‘Bouchon’ and not ‘Ouchon’

Answer: I am so sorry for the mistake. “Ouchon” was replaced by “Bouchon” in the revision.

Q4: Line 95 and 97 – bold or italicise.

Answer: Thanks for your suggestion. It had been revised to bold in the revision.

Q5: Equation 3 – Define C1

Answer: Thanks for your suggestion. c1 refers to the concentration of Sudan red staining solution in Soxhlet extraction oil. We had added this sentence in the revision.

Q6: Line 125 - This should come before the frying process

Answer: Thanks for your suggestion. The Sudan red staining experiment was supplemented in 2.2 and combined with the frying experiment.

Q7: Line 142 – baked or fried?

Answer: I am so sorry for the poor expression. It was modified to “The reacted hydrolyzate of hydrochloric acid (1mL) was dried in a DHG-9240A oven (Shanghai Zhuohao Laboratory Equipment Co., Ltd., Shanghai, China) at 60°C”.

Q8: Line 212 – oil penetration chart not mentioned in the Figure legend.

Answer: Thanks for your suggestion. “A indicates the Sudan red staining of sample. a, b, c, d and e indicate the images of samples under 0%KGM, 0.4%KGM, 0.8%KGM, 1.2%KGM and 1.6%KGM, respectively.” was added in the revision.

Q9: Line 239 - the one absorbed during cooling

Answer: Thanks for your suggestion. It was replaced by “It was showed that the oil content absorbed during frying was significantly lower than the one absorbed during cooling (p < 0.05).”.

Q10: Lines 248-249: If the inner surface was smooth as seen in the microstructure, how come the pore equivalent diameter is higher? One would think the rougher outer surface should possess a higher pore equivalent diameter. But that is not the case in the results. Please explain.

Answer: Thanks for your suggestion. The calculation of the pore equivalent diameter was performed with ImageJ software according to the method of Zhu et al. [20]. After the microstructure chart was imported into the software, the software would automatically identify the pores on the chart and calculate the pore equivalent diameter. The roughness of the outer surface might cause a significant error in the calculation results. After careful discussion, we decided to delete the data of the pore equivalent diameter of the outer surface. Combined with the article, deleting this set of data has little effect on the analysis. Thank you again.

Q11: Line 256 - The statement is not clear. Is this a proposed solution to counter the effect of oil increase with increasing KGM?

Answer: Thanks for your suggestion. This sentence was to confirm our speculation by citing literature. For a more accurate representation, it had been modified to “Therefore, we thought that 0.8% was the appropriate addition amount of KGM in this experiment. Lumanlan et al. [15] found that adding an appropriate amount of hydrocolloids such as 0.5% xanthan gum and 0.5% guar gum into the batter would form a smooth and compact crust to prevent oil from infiltrating.”.

Q12: Line 283 - Which previous research? Please cite.

Answer: Thanks for your suggestion. This should be a misrepresentation. “the previous research” was replaced by “the above research results” in the revision.

Q13: Figures 5b and c – the standard deviation is quite high meaning the triplicate data were spread out rather than clustered as other results. The coefficient of variation for the two datasets should be determined. If it is less than 20%, the data should be accepted. If not, the data should be re-calculated.

Answer: Thanks for your suggestion. We had calculated that the coefficient of variation of the original data was over 20%. We set up three groups of parallel experiments during our experiment, but one of them had a significant error. We re-analyzed the data and decided to remove the set of data with significant errors, and recalculated for the remaining two sets of data. Combined with other data analysis, the recalculated data has little effect on the overall trend. Thank you again.

Reviewer 3 Report

The title would look better if instead of reducing the oil content, one would talk about oil gain or oil absorption.
KGM is mentioned in the abstract, it would be convenient to define it in the abstract itself and then use the abbreviation KGM (Konjac Glucomannan) which is only defined on page 2.
The abstract could be improved by briefly mentioning the experiments carried out.
It is necessary to clarify what is meant by direct frying, frying is classified into surface frying and dip frying....
In materials and methods, grass carp is mentioned, my question is if there is any quality specification to characterise this raw material?
In materials and methods there is a reference to "other accessories", is it necessary to specify what it refers to?
It is necessary to review and improve the wording of point 2.2 and also to review the general wording of the document.
The frying process is not specified in terms of the equipment used, volume of oil, origin of the oil, oil/product ratio, etc.
In some parts of the manuscript the equipment used is detailed but this is not the case for all equipment/analyses, it is necessary to standardize.
It is not clear how the experiments were carried out in 2.4 it talks about another fryer? it is necessary to detail well how the frying experiments were carried out.
Review the explanation/mention of all variables in the equations used.
Revise the wording of point 2.5 is not clear.
It is important that the methods are correctly described and referenced.
The method for microscopic observations is not correctly specified.
Not enough detail on how the crust was removed.
Point 2.9 is not clear, the description needs to be improved.
2.12. does not specify the version of the software used.
It is not clear what is the contribution/relevance of figure 1 battering rate.
In 3.2 viscosity is mentioned, however, it was not measured, needs to be revised.
In the oil distribution results, the areas could be measured and quantitative information could be obtained.
It is necessary to revise the discussion, it is out of order.
Avoid using words like "It was speculated".
It is not clear how the pore diameter was measured, could the porosity of the sample also have been calculated?
In point 3.8 which is supposed to be the discussion only mentions the diagram (Figure 6) ...which otherwise would be good for a graphical abstract but I don't know if the place it is in the manuscript is a contribution.

The conclusions could be improved and also include possible practical applications based on the findings of the paper.

Author Response

Dear Reviewer,

Thank you for reviewing our manuscript and constructive suggestions concerning our manuscript entitled “Effects of Konjac glucomannan on oil absorption and safety hazard factor formation of fried battered fish nuggets” to Foods (ID: foods-1703084). These suggestions would help us to improve the quality of the paper, as well as the important guiding significance to our researches. We have studied comments carefully and have made correction. Revised portion are marked in red in the paper. The main corrections in the paper and the response to the reviewer’s comments are as follows.

Response to the reviewer’s comments,

Q1: The title would look better if instead of reducing the oil content, one would talk about oil gain or oil absorption.

Answer: Thanks for the valuable suggestion. The title was replaced by “Effects of Konjac glucomannan on oil absorption and safety hazard factor formation of fried battered fish nuggets”.

Q2: KGM is mentioned in the abstract, it would be convenient to define it in the abstract itself and then use the abbreviation KGM (Konjac Glucomannan) which is only defined on page 2.

Answer: Thanks for your suggestion. The definition had been revised in the Line 12.

Q3: The abstract could be improved by briefly mentioning the experiments carried out.

Answer: Thanks for the valuable suggestion. “The purpose of this study was to investigate the effects of konjac glucomannan (KGM) on oil absorption and formation of safety hazard factors in fried battered fish nuggets by measuring advanced glycation end products (AGEs) and acrylamide contents. Other physicochemical properties were determined to explore the reason for oil absorption and formation of safety hazard factors.” was modified in the abstract.

Q4: It is necessary to clarify what is meant by direct frying, frying is classified into surface frying and dip frying....

Answer: Thanks for your suggestion. Our frying experiments were both deep fried. The batter frying refers that the fish nuggets were battered followed by deep fried. Direct frying refers deep frying the fish nuggets directly. In order to distinguish between the two deep frying, we chose to express them in terms of direct frying and batter frying.

Q5: In materials and methods, grass carp is mentioned, my question is if there is any quality specification to characterise this raw material?

Answer: Thank you very much. We can guarantee that the grass carp used in each experiment is the live grass carp purchased from Huazhong Agricultural University market (Wuhan, Hubei, China), and the quality of grass carp is controlled within 2.5-3.0kg.

Q6: In materials and methods there is a reference to "other accessories", is it necessary to specify what it refers to?

Answer: Thanks for your suggestion. Other accessories include excipients such as salt and baking powder and chemicals such as petroleum ether (30-60℃). They were added in the 2.1.

Q7: It is necessary to review and improve the wording of point 2.2 and also to review the general wording of the document.

Answer: Thanks for your suggestion. The wording of point 2.2 had been modified to follow the order of the frying experiment in the revision.

Q8: The frying process is not specified in terms of the equipment used, volume of oil, origin of the oil, oil/product ratio, etc.

Answer: Thanks for your suggestion. The DF-6L fryer was purchased from Guangdong Jieguan Co., Ltd (Guangdong, China). Rapeseed oil was purchased from Shandong Luhua Co., Ltd (Shandong, China). Battered fish nuggets (6 pieces) were put into the fryer (containing 4L fresh rapeseed oil). These were supplemented in the revision.

Q9: In some parts of the manuscript the equipment used is detailed but this is not the case for all equipment/analyses, it is necessary to standardize.

Answer: Thanks for your suggestion. The equipment had been standardized in the revision.

Q10: It is not clear how the experiments were carried out in 2.4 it talks about another fryer? it is necessary to detail well how the frying experiments were carried out.

Answer: Thanks for your suggestion. Two DF-6L fryers (Guangdong Jieguan Co., Ltd., Guangdong, China) were used to perform frying experiment which was carried out at 180±2°C (measured by an AT-380 infrared thermometer (Guang Dong Ma Co., Ltd., Guangdong, China), among which one fryer containing Sudan Red B (0.75g/L) that was dissolved in rapeseed oil at 60 °C for 4 h, and another fryer containing rapeseed oil only. The battered fish nuggets were firstly fried in the rapeseed oil only fryer and then quickly (10 s before ending frying) moved to the Sudan Red B containing fryer.

Q11: Review the explanation/mention of all variables in the equations used.

Answer: Thanks for your suggestion. We had made corresponding additions in the revision.

Q12: Revise the wording of point 2.5 is not clear.

Answer: Thanks for your suggestion. The 2.5 had been modified in the revision to “The fish nuggets before battering were weighed by BL-2200H percent electronic balance (Zhejiang Baijie Instrument Co., Ltd., Zhejiang, China) as m1 (g). The battered fish nuggets were accurately weighed as m2 (g).”.

Q13: It is important that the methods are correctly described and referenced.

Answer: Thanks for your suggestion. The methods such as 2.2, 2.3, 2.5, 2.7, etc. had been correctly described and referenced in the revision.

Q14: The method for microscopic observations is not correctly specified.

Answer: Thanks for your suggestion. The method of microscopic observations was modified to “The microstructure was observed using the method of Zeng et al. [23] with slight modification. The crusts of the samples were cut into slices of about 2 mm x 2 mm x 1 mm with a ruler and a scalpel. The slices were fixed with 2.5% glutaraldehyde, and dehydrated sequentially with gradient ethanol solutions for 10 min each. The slices were immersed for 30min with isoamyl acetate. Finally, the slices were freeze-dried. The slices were glued to the stage using conductive glue and the surface was coated with gold film. Microstructural characteristics in outer and inner surface of slices were observed using a JSM-6390LV scanning electron microscope (Tianmei (China) Scientific Instrument Co., Ltd., Beijing, China) at 1000× and 3000× magnification.”.

Q15: Not enough detail on how the crust was removed.

Answer: Thanks for your suggestion. “After frying, the samples were naturally cooled to room temperature. Next, the crust and inner layer of the battered fried fish nuggets were separated manually with a scalpel.” was added in the revision in the 2.2.

Q16: Point 2.9 is not clear, the description needs to be improved.

Answer: Thanks for your suggestion. It had been modified in the revision.

Q17: 2.12. does not specify the version of the software used.

Answer: Thanks for your suggestion. The 2.12 had been modified in the revision to “Excel 2019 software was used for data statistics. Origin 2020 software was used for graphing. SPSS 25.0 was used for significant analysis”.

Q18: It is not clear what is the contribution/relevance of figure 1 battering rate.

Answer: Thanks for your suggestion. The main reasons for determination of the battering rate were as follows: 1) Battering rate could affect product yield. 2) KGM could absorb water. It would affect the thick of the batter with different KGM addition. The battering rate could reflect the thick of the batter on the side. It corresponded to the thickness of the crust in the Sudan red staining chart in Figure 2. 3) A high battering rate indicated that the batter was not uniform. It had an impact on oil absorption, which could affect the generation of safety hazard factors. Therefore, we measured the battering rate.

Q19: In 3.2 viscosity is mentioned, however, it was not measured, needs to be revised.

Answer: Thanks for your suggestion. It was modified by “KGM could absorb water 80-100 times its own weight, and the thick of its aqueous solution increased with the increase of KGM addition. [30]. It was most likely that higher KGM addition could absorb more free water, making the batter difficult to drip, thereby increasing battering rate and product yield.”.

Q20: In the oil distribution results, the areas could be measured and quantitative information could be obtained.

Answer: Thanks for your suggestion. The fish nuggets had a fixed size (3.0×3.0×1.5 cm). Therefore, there was little difference in the surface area of the fried battered fish nuggets. We chose to measure the oil penetration depth to correspond to the previously determined oil content.

Q21: It is necessary to revise the discussion, it is out of order.

Answer: Thanks for the valuable suggestion. The discussion had been modified in the revision. In the discussion, we mainly discussed the reason why KGM inhibited the production of AGEs and acrylamide.

Q22: Avoid using words like "It was speculated".

Answer: Thanks for your suggestion. We carefully checked and modified these sentences and marked them in red in the revision.

Q23: It is not clear how the pore diameter was measured, could the porosity of the sample also have been calculated?

Answer: Thanks for your suggestion. The calculation of the pore equivalent diameter was performed with ImageJ software according to the method of Zhu et al. [20]. After the microstructure chart was imported into the software, the software would automatically identify the pores on the chart and calculate the pore equivalent diameter. As the microstructure chart was a plan view, it was not possible to measure the depth of the pores and calculate the porosity. We'll look for other ways to measure pore depth later. Thank you very much.

Q24: In point 3.8 which is supposed to be the discussion only mentions the diagram (Figure 6) ...which otherwise would be good for a graphical abstract but I don't know if the place it is in the manuscript is a contribution.

Answer: Thanks for the suggestion. Thanks for the suggestion. Figure 6 was mainly a summary of the full text. After careful discussion, we thought figure 6 was more like a graphical abstract. Therefore, we decided to delete it.  

Q25: The conclusions could be improved and also include possible practical applications based on the findings of the paper.

Answer: Thanks for the suggestion. “Based on the experimental results, 0.8% KGM could be added in batter in order to de-press oil absorption and formation of AGEs and acrylamide in fried battered foods.” was added to the conclusion. Thank you again.

Round 2

Reviewer 1 Report

The authors have addressed most of my concerns. There is one specific suggestion I hope authors can consider.

The authors used different units (wet basis, dry basis) to express moisture content and oil content, respectively. And they were in percentage. For better comparsions, please use g/g or g/100g in dry basis for both mositure and oil content.

Author Response

Dear Reviewer,

Thank you for reviewing our revision and constructive suggestions concerning our revision entitled “Effects of Konjac glucomannan on oil absorption and safety hazard factor formation of fried battered fish nuggets” to Foods (ID: foods-1703084). These suggestions would help us to improve the quality of the paper, as well as the important guiding significance to our researches. We have studied comments carefully and have made correction. The main corrections in the paper and the response to the reviewer’s comments are as follows.

Response to the reviewer’s comments,

Q1: The authors used different units (wet basis, dry basis) to express moisture content and oil content, respectively. And they were in percentage. For better comparisons, please use g/g or g/100g in dry basis for both moisture and oil content.

Answer: Thanks for the valuable suggestion. We changed the units of moisture and oil contents to g/g dry basis. “The moisture and oil contents in the sample were expressed as g/g dry basis.” was added in the 2.3. The chart changes were shown in Figure 1 A, B. Thank you very much.

Reviewer 3 Report

The article improved markedly after the first revision.

Author Response

Dear Reviewer,

        Thank you very much for the careful review and affirmation. These suggestions would help us to improve the quality of the paper, as well as the important guiding significance to our researches. Thank you again.